# What's in a Name? Patterns, Trends, and Suggestions for Defining Non-Perennial Rivers and Streams

**Michelle H. Busch** [1,*], **Katie H. Costigan** [2], **Ken M. Fritz** [3], **Thibault Datry** [4], **Corey A. Krabbenhoft** [5], **John C. Hammond** [6,†], **Margaret Zimmer** [7], **Julian D. Olden** [8], **Ryan M. Burrows** [9,10], **Walter K. Dodds** [11], **Kate S. Boersma** [12], **Margaret Shanafield** [13], **Stephanie K. Kampf** [14], **Meryl C. Mims** [15], **Michael T. Bogan** [16], **Adam S. Ward** [17], **Mariana Perez Rocha** [18], **Sarah Godsey** [19], **George H. Allen** [20], **Joanna R. Blaszczak** [21], **C. Nathan Jones** [22] and **Daniel C. Allen** [1]

1 Department of Biology, University of Oklahoma, Norman, OK 73072, USA; dcallen@ou.edu
2 School of Geosciences, University of Louisiana, Lafayette, LA 70504, USA; katie.costigan@louisiana.edu
3 Office of Research and Development, U.S. Environmental Protection Agency, Cincinnati, OH 45220, USA; fritz.ken@epa.gov
4 INRAE, UR RiverLY, Centre Lyon-Villeurbanne, CEDEX 69100 Villeurbanne, France; thibault.datry@inrae.fr
5 College of Arts and Sciences and RENEW Institute, University at Buffalo, Buffalo, NY 14228, USA; ckrabben@buffalo.edu
6 Department of Ecosystem Science and Sustainability, Colorado State University, Fort Collins, CO 80526, USA; john.christopher.hammond@gmail.com
7 Department of Earth and Planetary Sciences, University of California, Santa Cruz, CA 95064, USA; margaret.zimmer@ucsc.edu
8 School of Aquatic and Fishery Sciences, University of Washington, Seattle, WA 98105, USA; olden@uw.edu
9 School of Ecosystem and Forest Sciences, The University of Melbourne, Burnley, VIC 3010, Australia; ryan.burrows@unimelb.edu.au
10 Melbourne Water, Docklands 3005, VIC 3008, Australia
11 Division of Biology, Kansas State University, Manhattan, KS 66506, USA; wkdodds@ksu.edu
12 Department of Biology, University of San Diego, San Diego, CA 92110, USA; kateboersma@sandiego.edu
13 College of Science and Engineering, Flinders University, Adelaide 5042, Australia; margaret.shanafield@flinders.edu.au
14 Ecosystem Science and Sustainability, Colorado State University, Fort Collins, CO 80526, USA; stephanie.kampf@colostate.edu
15 Department of Biological Sciences, Virginia Tech, Blacksburg, VA 24060, USA; mims@vt.edu
16 School of Natural Resources and the Environment, University of Arizona, Tucson, AZ 85721, USA; mbogan@arizona.edu
17 O'Neill School of Public and Environmental Affairs, Indiana University, Bloomington, IN 47405, USA; adamward@indiana.edu
18 CAPES Foundation, Ministry of Education of Brazil, Brasilia 70297, Brazil; mperezrocha@gmail.com
19 Department of Geosciences, Idaho State University, Pocatello, ID 83209, USA; godsey@isu.edu
20 Department of Geography, Texas A&M University, College Station, TX 77843, USA; geoallen@tamu.edu
21 Department of Natural Resources and Environmental Science, University of Nevada, Reno, NV 89557, USA; jblaszczak@unr.edu
22 Department of Biological Science, University of Alabama, Tuscaloosa, AL 35401, USA; cnjones7@ua.edu
* Correspondence: buschmh@ou.edu or mhopebusch@gmail.com
† Now at U.S. Geological Survey, MD-DE-DC Water Science Center, Catonsville, MD 21228, USA.

**Abstract:** Rivers that cease to flow are globally prevalent. Although many epithets have been used for these rivers, a consensus on terminology has not yet been reached. Doing so would facilitate a marked increase in interdisciplinary interest as well as critical need for clear regulations. Here we reviewed literature from Web of Science database searches of 12 epithets to learn (Objective 1—O1) if epithet topics are consistent across Web of Science categories using latent Dirichlet allocation topic

modeling. We also analyzed publication rates and topics over time to (O2) assess changes in epithet use. We compiled literature definitions to (O3) identify how epithets have been delineated and, lastly, suggest universal terms and definitions. We found a lack of consensus in epithet use between and among various fields. We also found that epithet usage has changed over time, as research focus has shifted from description to modeling. We conclude that multiple epithets are redundant. We offer specific definitions for three epithets (non-perennial, intermittent, and ephemeral) to guide consensus on epithet use. Limiting the number of epithets used in non-perennial river research can facilitate more effective communication among research fields and provide clear guidelines for writing regulatory documents.

**Keywords:** non-perennial; intermittent; ephemeral; temporary; stream; river; literature review; latent Dirichlet allocation; text mining; synthesis

## 1. Introduction

Rivers and streams that cease to flow at some point in time or space—hereafter referred to as non-perennial—are hydrologically diverse and globally prevalent. Their defining characteristic is a temporary lack of surface flow, which leads to isolated pools or dry channels [1–3]. By some estimates, these watercourses comprise at least 30% of the global river network [4], reaching up to 44% of the river network in South Africa [2], 59% in the United States [5], and about 70% of river channels in Australia [6]. The occurrence of non-perennial rivers is expected to increase in the future due to shifts in global climate, construction of dams and impoundments, and increased water abstractions to meet human demands [6–10]. Despite their prevalence worldwide, our understanding of the hydrology and ecology of non-perennial rivers is minimal when compared to perennial systems [1,6]. While non-perennial river systems have recently become the focus for researchers across many fields, the terminology used to describe them is incredibly diverse [11].

A number of terms currently exist to describe the temporal and spatial characterization of surface flow cessation or the partial or complete drying of rivers and streams (hereafter referred to as epithets, e.g., intermittent, ephemeral, temporary, dry, etc.) [1–3]. The issue of using multiple terms to describe the same concept has plagued ecology for years and is prevalent in some of the most frequently used terms in ecology [12–14] and hydrology [15]. A recent survey found substantial differences in how laypeople and experts consider words like river, dam, and river basin [13]. These examples demonstrate the confusion and misunderstanding that surround word use and scientific concepts, preventing clear and effective communication and potentially hindering scientific progress and effective management [15].

Standardized terminology is required to unify the rapidly growing literature on non-perennial streams and rivers [1] because effective science communication is "dependent on careful definition" [16]. Previous attempts to define non-perennial rivers exist, though their definitions have not been widely accepted [2,17,18]. Uys and O'Keeffe [2], for example, provide detailed terms and definitions based on various hydrologic stages and yearly timing of intermittency, though their paper is primarily focused on rivers and streams in South Africa. A more recent attempt focused on the timing and location of aquatic habitats, such as riffles and pools [17]. Despite these efforts, there is no global consensus on terminology [1]. It has been theorized that this could be in part because a single river may fit into various categories depending on the section of the channel requiring a definition [19]. Previous attempts to define non-perennial terminology also rely on specific details about the timing and magnitude of flows of the river in question [2,17]. These details may not always be available, however, as hydrologic data on non-perennial rivers and streams are scarce due to the lack of river gages on them and the historical research focus on perennial rivers [1,19,20].

Despite the lack of current consensus, using common terms and definitions ensures continued opportunities for multidisciplinary research [11], supports research synthesis such as comprehensive reviews and meta-analyses [21], and ensures that assumptions behind the definition of non-perennial waters are explicit [11]. Taken together, clarifications on terminology should further aid progress in connecting research and management [11]. For example, in the United States, the Clean Water Act allows states to adopt individual definitions and water quality standards after approval from the Environmental Protection Agency. These state-by-state definitions of non-perennial rivers, along with a lack of consistent epithets, allow for a wide variety of policy and enforcement. Out of 56 states and territories, only 17 define "ephemeral" and 20 define "intermittent" waterways. This lack of consistency allows for protections of these valuable resources to vary widely, placing non-perennial systems further at risk and complicating management efforts [22,23]. Common nomenclature will allow for easier comparisons across fields, lead to more inter- and multidisciplinary research, [21] and be inclusive of the wide hydrologic range of non-perennial systems [1,24]. More open and connected research will also help refine and build conceptual models and meta-analyses [21], allowing for clearer research synthesis and better protection for non-perennial river systems.

Our aim in this review was to identify, define, and synthesize the diversity of epithets used to describe non-perennial streams in research [1–3] across various fields and to provide common terminology [1]. Here we address the nomenclature surrounding non-perennial river systems and review how their various epithets are used. Our objectives are to: (Objective 1, hereafter O1) synthesize the large and growing literature to evaluate how epithets are used across Web of Science categories; (O2) assess how epithet use has changed over time; and (O3) identify how epithets have been defined throughout the literature. We use bibliometric techniques including topic modeling to explore how epithets for non-perennial river systems vary across research fields (e.g., hydrology, ecology) and over time to allow for a consensus in epithet use. In this way, we provide an overview of non-perennial river research trends to allow for more consistent terminology to further inter- and multidisciplinary non-perennial river research and protections. Finally, we suggest universal terms and definitions to provide common epithets across research fields.

## 2. Materials and Methods

### 2.1. Data Sources

We performed a comprehensive literature review using the search engine in Clarivate Web of Science (WoS; Core Collection) for papers published between 1 January, 1900 and 30 May, 2019. Our searches were not limited to the English language, though all analyses assumed that text was written in English. WoS has indexed scholarly books, peer-reviewed journals, reviews, editorials, chronologies, and reports within multiple research fields since 1900. We created two-word search phrases that included (1) an epithet (a common descriptor used to describe a non-perennial system) and (2) a waterbody term, a noun for a flowing (i.e., lotic) freshwater system. We first generated a list of epithets of non-flowing conditions by selecting terms found in previous reviews of non-perennial literature [1–3,24] and those resulting from discussions among authors for a total of 12 epithets (Table 1). We completed separate searches for "semi-perennial" and "non-perennial" systems due to the large number of papers that refer to perennial systems (i.e., those with uninterrupted surface flow). After conducting the semi-perennial and non-perennial searches separately, we combined the results to create the final "non-perennial" corpus (a corpus is defined as a set of text documents which are analyzed together). We used the same approach to create the "non-permanent" corpus, again to exclude papers that focus on perennial or permanent streams.

Next, we paired each epithet (Table 1) with an adjacent waterbody term [1,3,25]. Similar to the epithets, waterbody terms were those commonly used to describe lotic systems and selected based on a series of discussions among authors. We included papers using both singular and plural terms (e.g., "river" and "rivers"). While we included 41 lotic waterbody terms, closer examination of search

results indicated that some lentic (still or non-flowing water) focused papers were included in our results. Lastly, we limited the search results according to 37 WoS research field categories that include journals publishing natural science research (see S1 for example search), resulting in 11,696 papers to be analyzed (Table 1).

**Table 1.** Table of search results from Web of Science. Epithets used to define non-perennial systems in the present study, paired with the source it was pulled from in brackets. We limited our search results by WoS categories to narrow our results for papers specifically related to natural river systems.

| Epithet and [Reference] | Number of Papers |
| --- | --- |
| Arid [4] | 837 |
| Discontinuous [5] | 223 |
| Dry [5] | 1580 |
| Ephemeral [5] | 1652 |
| Episodic [6] | 201 |
| Intermittent [5] | 1582 |
| Interrupted [5] | 103 |
| Irregular [26] | 347 |
| Non-Perennial [5] | 59 |
| Non-Permanent [5] | 40 |
| Seasonal [5] | 4358 |
| Temporary [5] | 1404 |
| Total | 11,696 |

We downloaded full search results for each epithet separately (includes author list, year published, WoS categories, abstracts) and combined text from the abstracts to create a unique, epithet specific final corpus (Table 1). Abstracts were decomposed into a list of individual words (tokenized) without punctuation or numbers. Each word was then reduced to its base form (stemmed; e.g., "rivers" and "riverine" both became "river") and common English words (stop words; e.g., "the", "and", "or") were removed. All analyses were run in R as described in the following sections [27]. A methodological flow chart was created to provide additional clarity (Figure S1).

*2.2. O1: Topical Differences among Epithets*

We compared the proportion of search results across categories as defined by WoS to quantify epithet use across fields (Figure 1). We then used a Fisher's exact test to determine if there were any significant differences between WoS categories and the number of papers per epithet. Due to the size of the matrix, we had to use a simulated $p$ value based on over a million replicates. We then deployed topic modeling with Bayesian latent Dirichlet allocation (LDA) models in the R package "textmineR" [28]. This model assumes that each document in a corpus is made up of multiple topics, and each topic is made up of multiple words. The model finds common clusters of words within documents, grouping them together and forming topics. For example, the topic "agriculture" is created due to the common occurrence of the following words found together throughout documents: soil *, water *, irrig *, yield *, us *, season *, crop *, and so on (* indicates the result of stemming). Using these clusters, the model calculates the prevalence of each topic within individual documents within the corpus and estimates the probability of finding each topic within an individual document (theta matrix). This analysis requires the number of topics to be decided a priori. We calculated and averaged the probabilistic coherence, which tests how understandable each topic is, for 1–50 topics to determine the number of topics for modeling [29]. We tested 1–50 topics to determine the trends of increasing the number of topics while keeping a smaller number of topics that would be manageable to explore. We chose six topics for topic modeling based on a plot of coherence versus the number of topics (Table A1; Figure S2).

We combined all 12 abstract corpora into a complete corpus for topic modeling. We calculated the probability of finding a topic in each document in the corpus with the theta matrix using the package

"textmineR" [28]. After designating the six topics, we extracted the top 20 words used to define each topic for comparison across epithets, and named each topic based on the 20 words associated with it. We averaged the theta matrix for each epithet and performed an unconstrained ordination using non-metric multidimensional scaling (nMDS; using the R package "vegan" [22]) to assess similarities in topical overlap among papers categorized according to the epithets (Figure 2).

### 2.3. O2: Epithet Uses over Time

We conducted WoS searches based on ten time frames and explored how publishing rates of the epithets have changed over time. Prior to 1990, very few papers were published (or cataloged by WoS) on non-perennial systems [1]. We therefore used three 25-year and one 15-year timeframes between 1900 and 1990. After 1990, publications on non-perennial systems became more frequent. Thus, we created timeframes for periods of five years. We used a Fisher's exact test to test the differences between the number of papers for each epithet across the time frames.

We used the same topic modeling approach described above [28] on the combined time series corpus to investigate changes in topics over time in the literature. We selected nine topics based on coherence calculations (Figure S3) and visualized their prevalence over time.

### 2.4. O3: Epithet Definitions

We first selected papers where epithets paired with "river" or "stream" were most important to compare epithet definitions. To select these papers, we calculated the "term frequency—inverse document frequency" (*tf_idf*) across the abstracts found in each corpus [28] (Equation (1)).

$$tf\_idf = \frac{(f_a)}{(n)} + log(\frac{N_d}{N_a}), \tag{1}$$

where $f_a$ is the frequency that word "*a*" appears in a document, *n* is the total number of words in a document, $N_d$ is the number of documents in the corpus, and $N_a$ is the number of documents within the corpus containing word "*a*". The *tf_idf* describes the relative importance of a word in each document within a larger corpus. This value increases proportionally to how frequently the word appears in a document but is offset by how frequently the word is found within the full corpus (Equation (1)). We limited waterbody terms to either "river" or "stream" for definition analysis to ensure search results would be limited to lotic freshwater systems (for example, TS = (* arid * NEAR/0 (river * OR stream *) and WC = (listed S2)), where TS stands for search term, * indicates any prefix or suffix, NEAR/0 forces adjacency for the two search terms, and WC refers to Web of Science Category).

We selected a maximum of 50 papers for each epithet (25 using river and 25 using stream), totaling 374 papers used to collect definitions (Table S1). We manually searched each paper selected to record definitions of the epithets used. If other epithets were used in a paper, they were noted along with any definitions for analysis. Not every paper with a high tf_idf value included a definition, thus, we randomly selected an additional 25 papers from each corpus (not limited by waterbody term) to increase sample sizes. This led to a maximum of 75 and a minimum of 28 papers selected for definition mining (total of 672 papers used for definition analysis; Table S1). Before analysis, definitions were carefully reviewed to ensure proper spelling and to remove openings (e.g., "ephemeral streams are characterized by low flow during summer months" became "low flow during summer months").

Definition corpora were too small to allow for LDA topic modeling. Therefore, we developed a set of important themes common across non-perennial river literature related to water sources, predictability, time frame, climate, and various phases of drying to assess how epithet definitions overlapped (Table S2). We reviewed each definition to assess its match to each theme. The number of themes that fit each epithet definition was totaled and divided by the total number of definitions per epithet term available. In this way, we assessed the degree of overlap among themes to see which epithets had the most similar definitions. We visualized epithet similarities using nMDS ordination

with Euclidean distance. We used the R package 'vegan' [30] with the proportions of each theme in epithet definitions for the nMDS. After running the initial nMDS, we excluded two themes that lacked a large proportion of any definitions (water scarcity and phases of drying: no subsurface flow) and "irregular," which only had one definition.

We then compared how various research fields use the same epithet. Using our results from definition mining, we selected five epithets whose definition analysis papers were over 80% covering non-perennial river systems (non-perennial, 100%; ephemeral, 98%; temporary, 89%; intermittent, 88%; arid, 83%; Table 2). We chose to assign WoS defined categories by broader research fields (ecology, hydrology, and eco-hydrology; Table S3) to simplify comparisons. We reviewed definitions in each field to create a summary definition.

## 3. Results

### 3.1. O1: Topical Differences among Epithets

A wide range of research fields use highly varied terms to describe non-perennial systems. While most WoS categories contained papers that used each of the 12 epithets, there were some epithets that were more associated with specific categories (Fisher's Exact Test, simulated $p$-value > 0.001 with $1e^{05}$ replicates; Figure 1). Categories that were least likely to be about non-perennial rivers (biochemistry, physical/multidisciplinary/applied chemistry, computer science, materials science) had the greatest proportions of the epithets "arid," "dry," and "discontinuous". Categories that are likely related to lotic non-perennial systems (water resources, marine and freshwater biology, multidisciplinary geosciences, environmental sciences, ecology) each contained over 1000 papers. These categories mostly used "seasonal" and "ephemeral," though "seasonal" was the epithet which dominated most categories. Seven of the twelve epithets were dominate across the 40 categories which were made up of at least 50 papers. Despite these differences, no single category was limited to the use of one epithet. Each category includes papers from at least six epithets (Figure 1).

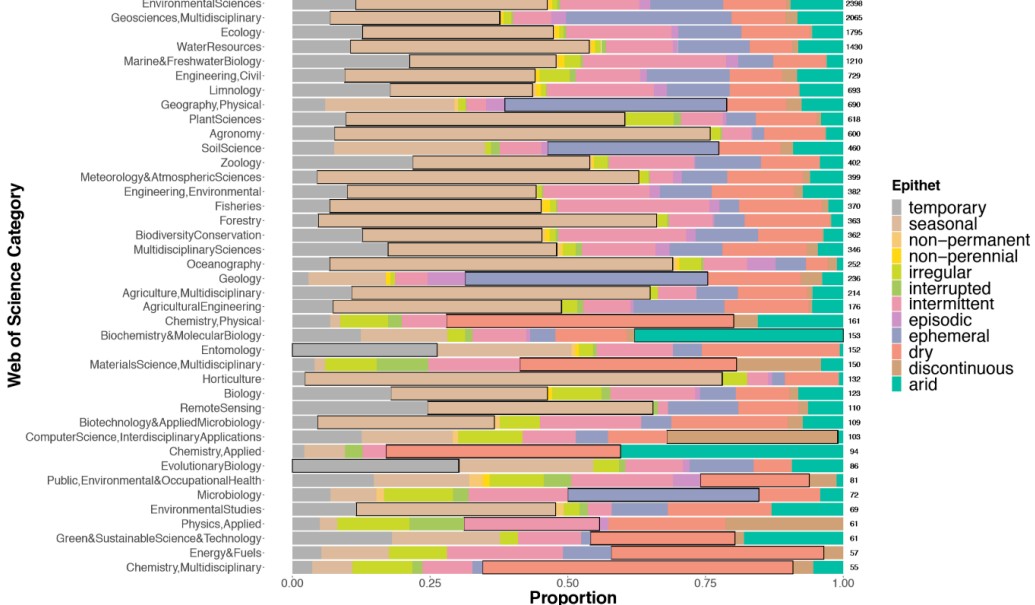

**Figure 1.** Proportion of epithets by Web of Science (WoS) category. Displayed are the 40 categories (out of 148 unique WoS categories) that contained at least 50 papers across all 12 epithet corpora on the *y*-axis. The proportion of each category made up of epithets is on the *x*-axis. Each color represents a different epithet, with the most dominate epithet per category highlighted in black. The total number of papers appearing in each category is displayed as the sample size along each column for reference (environmental sciences had the largest number with 2398 while multidisciplinary chemistry had the smallest with 55 papers).

Topic modeling from the complete corpus identified only six topics that we labeled based on the top 20 words associated with each topic (Appendix A). When exploring the similarities between epithets and topics, we found four main groups of epithets, each related to at least one topic (Figure 2). We calculated the stress of the nMDS to demonstrate how well the ordination works on a two-dimensional plane, with a stress below 0.2 demonstrating a good representation [31]. The largest cluster contains "arid," "dry," "intermittent," "non-permanent," "seasonal," and "temporary" and is associated with topics focused on agriculture, hydrology, and vegetation. The second largest cluster includes "discontinuous," "interrupted," and "irregular," though "discontinuous" is farther removed from the other epithets. The topic most associated with this cluster is temperature. "Non-perennial" was the smallest cluster, with a close association to ecohydrology.

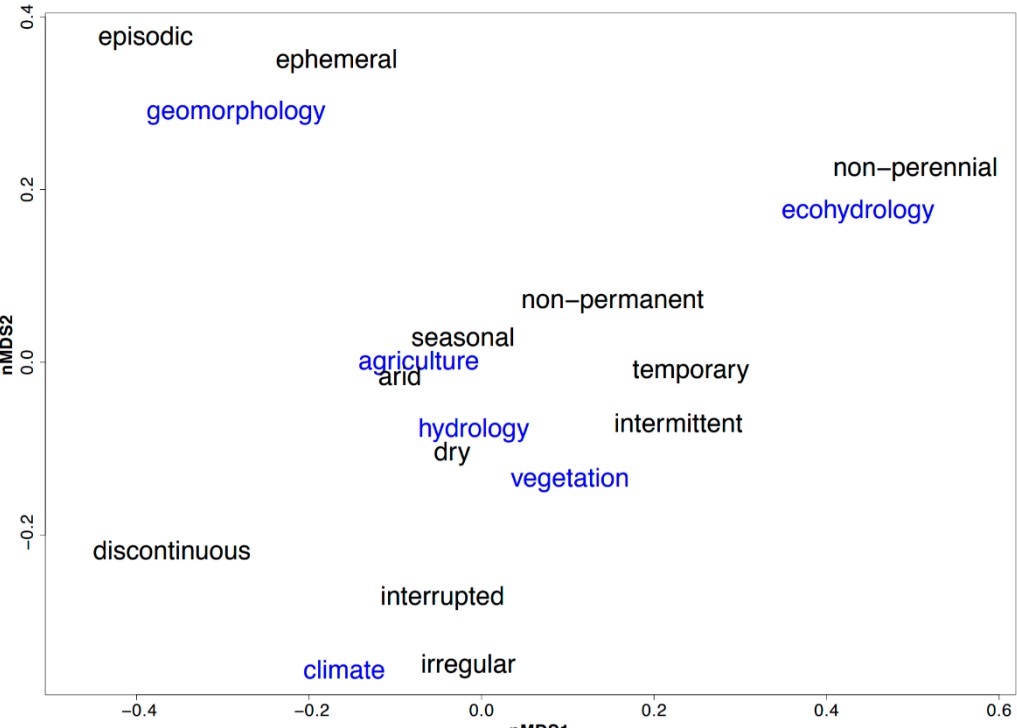

**Figure 2.** A non-metric multidimensional scaling (nMDS) ordination (stress = 0.1029) representing similarities among the six topics found from latent Dirichlet allocation (LDA) topic modeling (blue words) and how they relate to papers about each epithet (black words). The large sample size made plotting each paper too cluttered, so average probabilities of the topic appearing in papers across all twelve corpora were used to understand broad, topical terms between epithets.

### 3.2. O2: Epithet Uses over Time

The rate of publications on non-perennial rivers increased drastically after 1990 (from 16.1 papers published per year to 154.2; Figure 3A). Publishing rates continued to increase quickly over time until the last time frame, 2016 to 2019. However, this time frame was much smaller than previous ones due to the search date of May 2019 as opposed to December 2020, which would have made it a full 5-year interval.

We found a difference in the proportion of the most common epithet per year (Fisher's Exact Test, simulated *p*-value > 0.001 with $1e^{05}$ replicates). After 1950, 11 epithets are found throughout literature (excluding "non-perennial"; Figure 3B). "Non-perennial" appears between 1990 and 1995, disappears between 2001 and 2005, and appears again after 2006 with less frequency than the other terms. "Seasonal," "ephemeral," and "arid" appear frequently in all time frames after 1950. "Intermittent" is prominent in all time frames and has been used at approximately the same rate since 2000. Up until 1975, "discontinuous" was used frequently (33% between 1900 and 1925 and between 1926 and 1950),

however its use decreased after 1950 (1% between 2016 and 2019). "Episodic" appears after 1975, though it also does not make a large proportion of results. "Interrupted," "irregular," "non-perennial," and "non-permanent" are similarly used infrequently over time.

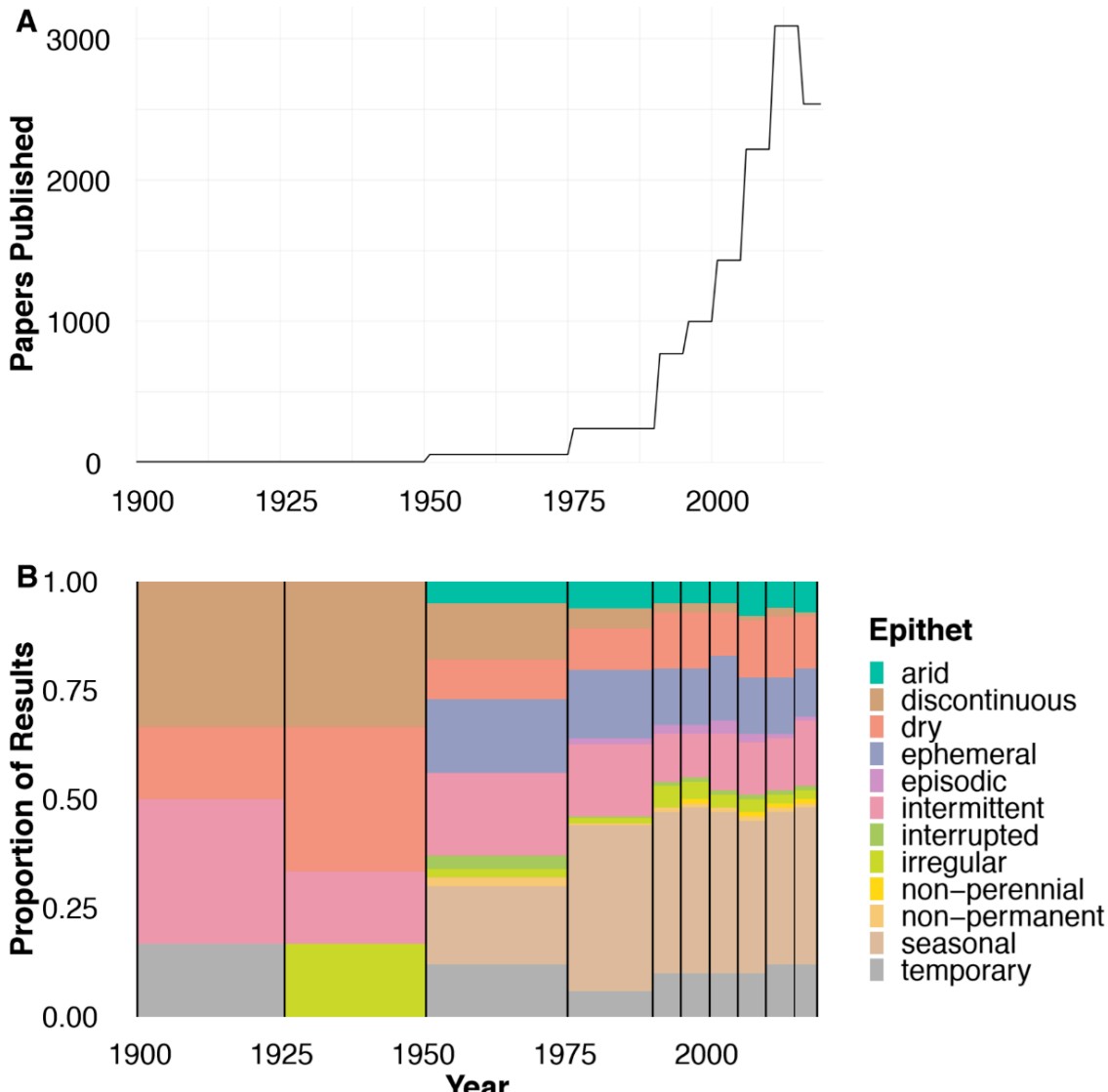

**Figure 3.** Number of total papers published (**A**) and proportion of papers published under different epithets (**B**) over time frames. Note the large jump in publication rates after 1990, and the steady increase until 2016. Papers from all 12 epithets are published between 1996 and 2000, though "non-perennial" does not appear in 2001–2005, all 12 are again represented in results from 2006 onward.

"Seasonal" has been used most frequently over time, followed by "dry" and "arid" (Figure 4). These three terms can also be related to the description of a study site, are broad, and are common in language and therefore may not require a specific definition. "Irregular," "episodic," "discontinuous," "interrupted," and "non-permanent" are used by the fewest papers over time.

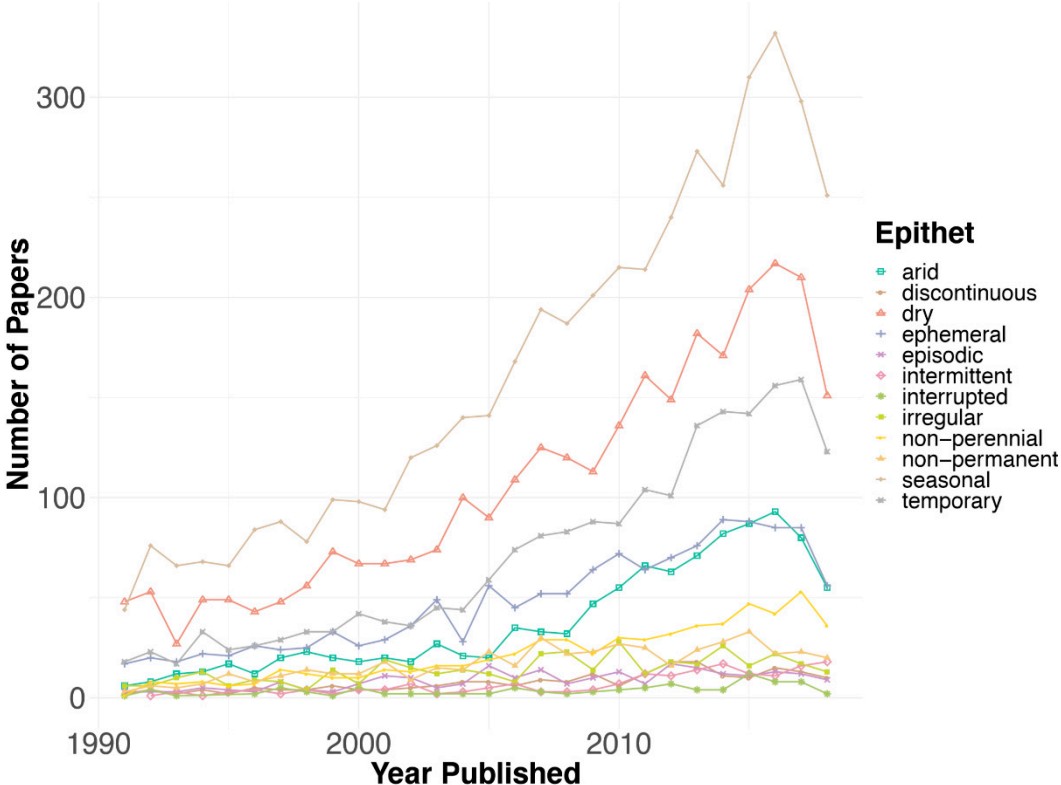

**Figure 4.** Publications of each epithet over time. Abstracts from 1990 to 2017 were compiled to see how epithet use has changed over time. Each epithet was counted once per abstract. Broader terms such as "seasonal," "dry," and "temporary" have been used the most frequently. "Temporary," "non-perennial," and "intermittent" are the three terms with the largest rate of increase over time.

Nine topics were identified using the LDA topic modeling from the combined Time Series Corpus. While some topics from the time frame sampling have the same names as those from epithet corpora, note they are from separate analyses (Table A2; Figure S4). Generally, the topics that have decreased in proportion over time are agriculture (9% decrease from 1991–1995 to 2016–2018) and geomorphology (6% decrease from 1991–1995 to 2016–2018). Topics that show the greatest increase over time are hydrology (14% increase from 1991–1995 to 2016–2018), community ecology (6% increase from 1996–2000 to 2016–2018), and modeling (5% increase from 1991–1995 to 2016–2018; Figure S3).

*3.3. O3: Epithet Definitions*

In total, 672 papers were reviewed for definition analysis. Although we limited our WoS searches, some search results included papers about topics other than non-perennial rivers or streams. For example, "dry stream" could be found in a chemistry paper testing adsorption ability of various volatile organic compounds under humid and dry streams of gas [32]. "Non-perennial" and "ephemeral" had the largest proportion of papers that were about non-perennial river systems (100% and 98.7%, respectively) while "episodic" and "irregular" had the smallest proportion of papers on non-perennial systems (19.1% and 11.1%, respectively; Table 3). Roughly 22% of "episodic" papers focused on episodic acidification in streams as opposed to drying flow regimes (e.g., [32,33]). "Non-perennial" was the smallest corpora with only 28 papers, though 100% of these papers were related to our topic (note that our WoS search for "non-perennial" and "non-permanent" excluded papers covering systems that flow continuously over time).

Out of the 672 papers analyzed for this study, 54% lacked any definition of its epithet. Out of the 452 papers classified to be about non-perennial river systems (67% of total papers explored), 39% did not include any definition (Table 2).

**Table 2.** Content of papers by epithet (NP refers to non-perennial). We examined 672 papers to mine definitions. However, many of these papers were not about non-perennial systems or lacked definitions. "Non-perennial" had the lowest number of papers included in the analysis, the largest percentage of papers that were about non-perennial systems, and one of the smallest proportions of papers without definitions (only "episodic" was lower).

| Epithet | Number of Papers | % About NP Systems | % of NP Papers Without Definitions | NP Papers Without Definitions | Final Definition Counts |
|---|---|---|---|---|---|
| Arid | 72 | 83.3% | 36.7% | 22 | 36 |
| Discontinuous | 36 | 33.3% | 66.7% | 8 | 4 |
| Dry | 75 | 66.7% | 52.0% | 26 | 24 |
| Ephemeral | 75 | 98.7% | 32.4% | 24 | 65 |
| Episodic | 63 | 19.1% | 16.7% | 2 | 13 |
| Intermittent | 75 | 88.0% | 33.3% | 22 | 59 |
| Interrupted | 33 | 33.3% | 54.6% | 6 | 5 |
| Irregular | 36 | 11.1% | 75.0% | 3 | 1 |
| Non-perennial | 28 | 100.0% | 17.9% | 5 | 23 |
| Non-permanent | 29 | 79.3% | 52.2% | 12 | 11 |
| Seasonal | 75 | 60.0% | 51.1% | 23 | 23 |
| Temporary | 75 | 89.3% | 31.3% | 21 | 51 |
| Total | 672 | | 38.5% | 174 | 315 |

In total, we compiled 315 definitions from the original 672 papers. "Irregular" was the only epithet that had one paper providing a definition, while "discontinuous" had four.

The themes identified for definition analysis were chosen to represent the broader ideas that characterize non-perennial river systems (Figure 5). "Intermittent," "temporary," and "non-perennial" overlap with one another. These epithets are strongly related to the topic "Phases of Drying: No Flow" (see Table S2 for full theme list). These three epithets are also related to "non-permanent" and "seasonal" and the themes "Phases of Drying: No Surface Water" and "Predictability/Seasonality." Conversely, "discontinuous" (with four definitions) was strongly related to the ideas of disconnections ("Phases of Drying: Low Flow" and Phases of Drying: Isolated Pools"). "Interrupted" was surrounded by various themes and yet not closely related to any ("Phases of Drying: Not Specific", "Source: Groundwater", and "Tied to Specific Landscape or Climate"). Based on the nMDS, "interrupted" and "discontinuous" are the least related to other epithets, yet also lack strong correlations with most themes. Generally, overlap exists among definitions found for all 12 epithets.

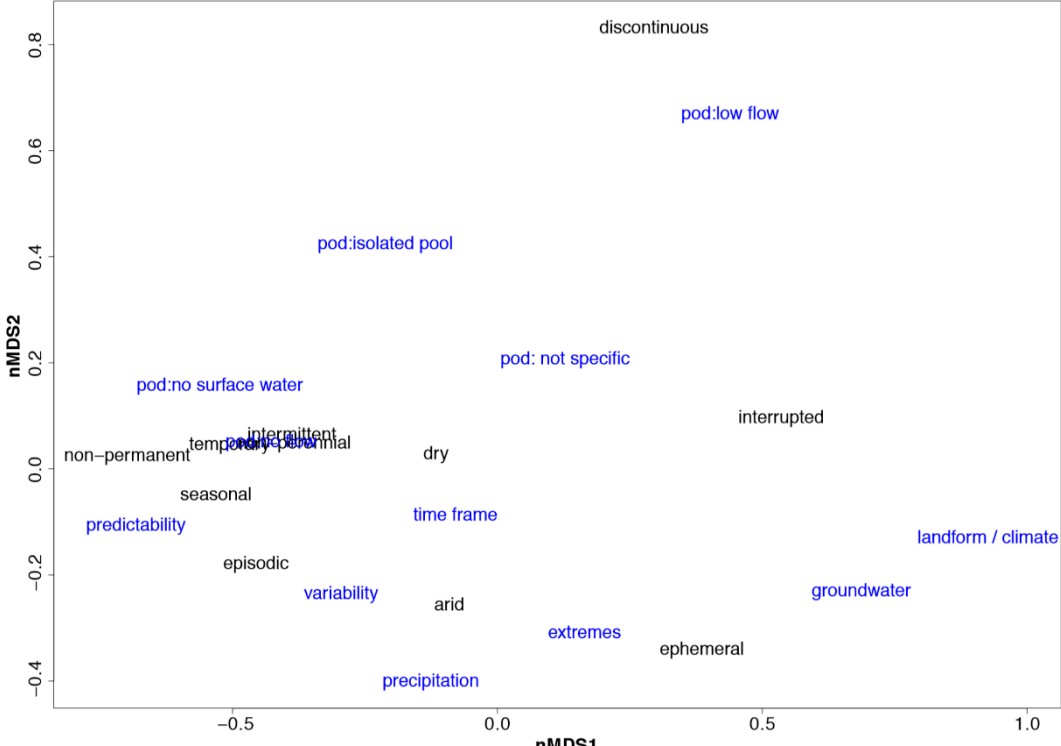

**Figure 5.** An nMDS representing the Euclidean distance among epithets and definition themes (stress = 0.0962). Epithets in blue text, definition themes in larger, black text. The section of the nMDS with overlap includes epithets "intermittent", "temporary", and "non-perennial" as well as the theme *Phases of Drying: No Flow.* Full theme list: predictability = predictability/seasonality; variability = variability/unpredictability; time frame = specific time frame mentioned; landform/climate = related to specific landform/climate; extremes = related to extremes (droughts and floods); pod = phases of drying: (Table S2).

We explored the five epithets that had the highest number of definitions to broadly examine how different fields use the same epithet (Table 3). The epithets we analyzed further were "non-perennial" (28 definitions), "ephemeral" (65 definitions), "temporary" (51 definitions), "intermittent" (59 definitions), and "arid" (72 definitions). Summary definitions were created for ecology, hydrology, and eco-hydrology fields (Table S3). These definitions also demonstrated overlap between epithets and among fields and many definitions were similar to one another.

**Table 3.** Summary definitions for arid, ephemeral, intermittent, non-perennial, and temporary by broad research fields. Each one of these five epithets had over 80% of their papers provide a definition.

| Epithet | Field | Summary Definition |
|---|---|---|
| Arid | Ecology | Natural drying and wetting phases following seasonal fluctuations, leads to natural expansion (connected stream) and contractions (isolated pools) |
| Arid | Hydrology | May not flow every year depending on precipitation, minimal groundwater recharge |
| Arid | Eco-Hydrology | Variable flows between or within years, sensitive to changes in climate |
| Ephemeral | Ecology | Streams that only flow variably for a short period of time, after precipitation events during certain times of the year |
| Ephemeral | Hydrology | Flow is scarce and sporadic in streams with high drainage, flow typically as a result of an extreme precipitation event |
| Ephemeral | Eco-Hydrology | Rivers without flow for most of the year, yet which have high intensity flooding periods in response to precipitation events |
| Intermittent | Ecology | Seasonal flowing and drying conditions that may result in isolated pools |

**Table 3.** *Cont.*

| Epithet | Field | Summary Definition |
|---|---|---|
| Intermittent | Hydrology | Regular wet and dry cycles with extreme floods and droughts, resulting in disconnected pools during the dry season |
| Intermittent | Eco-Hydrology | Naturally dynamic and variable cycles of wetting and drying that can change from year to year in response to precipitation patterns |
| Non-Perennial | Ecology | Lose surface water in drying and rewetting cycles for a period of time in most years |
| Non-Perennial | Hydrology | Variable low and no flow periods |
| Non-Perennial | Eco-Hydrology | Loss of flow and connectivity, reducing a stream to isolated pools during dry season |
| Temporary | Ecology | Rivers that cease to flow for a period of time during cycle of drying and rewetting |
| Temporary | Hydrology | Recurrent dry phase with no flow for variable time periods |
| Temporary | Eco-Hydrology | Rivers that experience wetting after precipitation events and drying in drier seasons leading to isolated pools |

## 4. Discussion

Consistent terminology remains fundamental to advance science and communicate its broader implications [1,14,21]. Here we highlight this issue with the diverse terminology used to describe non-perennial rivers systems over the past century [1–3]. As non-perennial river systems are both widespread and increasing numerically in response to climate change and increasing human demands [6–9], research on these systems continues to grow [1]. By conducting a bibliometric review of the published literature, we assessed whether epithet use is consistent across WoS categories using topic analysis; examined how epithet use to describe non-perennial rivers has changed over time; and identified how epithets have been defined. We conclude with recommendations for universal terms and definitions moving forward.

### 4.1. O1: Topical Differences among Epithets: Lack of Consensus

Our findings demonstrate the lack of consensus on epithet use within specific categories—each one included at least six epithets within their search results. While some epithets are more prevalent in certain categories than others (Fisher's Exact Test, simulated *p*-value > 0.001), many epithets are common across multiple categories. For example, "seasonal," "temporary," and "dry" can be found in each WoS category we explored (Figure 1). These epithets may simply be too broad, enabling their use across multiple categories.

We found widespread epithet use within and among WoS categories, with many of them using multiple epithets to describe non-perennial river systems. Despite limiting our searches to categories closely related to non-perennial river systems, some papers in our search results were in different WoS categories (e.g., computer science). Generally, these categories contained fewer papers than those likely related to freshwater lotic systems (e.g., water resources, environmental sciences, ecology, marine and freshwater biology). However, computer science, applied chemistry, and applied physics were all categories that had at least 50 papers across the 12 corpora. Epithets that made up the largest proportion in these categories were "arid," "dry," "discontinuous," and "irregular." For example "arid" makes up a large proportion of both the applied chemistry and biochemistry and molecular biology fields (e.g., [23]). Therefore, some epithets used in research surrounding non-perennial systems are also used for different fields beyond describing water flow in lotic ecosystems. While these papers could be related to non-perennial systems, the number of papers in our search results were too many to thoroughly examine as has been done in other LDA analyses [34]. This highlights the need for specific titles and keywords to ensure related papers appear in WoS searches.

While hydrology was not a distinct category included in the WoS searches, it did appear as a topic in LDA topic modeling. This topic was closely related to agriculture and vegetation themes and was

related to the largest cluster of epithets ("non-permanent," "seasonal," "arid," "dry," "intermittent," and "temporary"). The relationship between these topics demonstrates the similar research themes within hydrology and ecology, as we also relate topics of agriculture and vegetation broadly to ecology. The six topics that came out of LDA topic modeling further demonstrate the overlap between epithets in general, suggesting that the use of twelve epithets in non-perennial research is excessive.

*4.2. O2: Epithet Uses over Time: Shift towards Syntheses*

Epithet usage has been dynamic during the past 120 years (Fisher's Exact Test, simulated *p*-value > 0.001). "Seasonal" was the most common epithet of non-perennial rivers used in the literature across the entire time period (Figure 3) and was consistently used across WoS categories (Figure 1). Over time, the epithets that appear the most frequently tend to be more broad ("seasonal," "temporary," "dry") as well as more likely to be found in WoS categories other than those most likely to be about non-perennial systems (Figures 1 and 4). These epithets are likely most often used in WoS categories that focus on topics other than non-perennial systems. This observation was supported by our definition analyses. Based on the papers used for definition mining, less than 70% of papers that used "dry" and "seasonal" were about topics involving non-perennial river ecosystems (66% and 60%, respectively). Some papers used "interrupted stream" to describe airflow over chemicals [32] or "episodic river" to discuss infrequent river acidification [32,35,36]. Moving forward, it may be beneficial to avoid the use of general epithets of non-perennial systems and instead be more specific in terminology, so publications become more accessible and more likely to be included in future searches of non-perennial rivers.

Our analysis of how topics have changed over time offers further insight into temporal trends in epithet use (Figure 5). Topics that have decreased in use over time appear to be more descriptive and less about the river systems themselves. Conversely, topics related to hydrology, community ecology, and modeling have increased in prevalence (Figure 5). This shift is logical—observations and descriptions about systems are needed before hypotheses can be made and tested. The shift found in our results reflects this and demonstrates an increase in syntheses. This trend has been reported in other non-perennial river research, where historical papers tend to be descriptive while contemporary research is focused on, for example, the impacts of climate change (modeling) or how human interactions will impact communities (community ecology) [1,37].

We found a large shift in publication rates in our time frame analysis after 1990, which may be due to historical events that increased societal and environmental awareness of non-perennial rivers. The North American Benthological Society (NABS) held a Stream Solute Workshop in 1990 [38] which led to multiple papers, including some which were the first to explore the biogeochemistry of non-perennial rivers (e.g., [1,38,39]). The 2000 Water Framework Directive (WFD) in Europe encouraged research to include river condition assessments, as the classification of "waterbodies" determines river protection status including non-perennial streams [1]. Similarly, in 2007 Australia established the Commonwealth Water Act to ensure proper management of the Murray–Darling Basin to protect intermittent rivers [22]. The story of non-perennial rivers is complex in the United States. In 2001, the court case of Solid Waste Agency of Northern Cook County v. U.S., the U.S. Army Corps of Engineers focused on the definition of "jurisdictional waters" within the Clean Water Act. Lawyers for the Solid Waste Agency argued that the Army Corps of Engineers did not have jurisdiction over waterways not adjacent to open water, which often include non-perennial rivers [40]. In 2006, Rapanos v. United States case highlighted the need for clear definitions of non-perennial rivers and led to a large increase in non-perennial river research as scientists worked to demonstrate the importance of non-perennial rivers to navigable waters [1,22]. The definition of jurisdictional waters further evolved in 2015 with the Clean Water Act [22] and again in 2020 when the Navigable Waters Protection Rule was completed, which included definitions for non-perennial rivers. Changes in policies and regulations within the past two decades surrounding non-perennial river systems have brought these systems further into the public eye, encouraging scientific research. With additional policies and water

governance decisions regarding the definition and classification of non-perennial rivers, such as the 2020 U.S. Navigable Water Protection Rule, research on these systems will continue to grow.

### 4.3. O3: Epithet Definitions: The Issue of Overlap

Over one third (38.5%) of all papers did not offer a clear definition for the epithet of non-perennial rivers. Missing or vague definitions could be due in part to the increase in research and familiarity around these systems [21] or because resources on technical definitions are not well known [41]. The limitations of our WoS searches were also demonstrated while attempting to mine definitions. Out of the 672 papers selected, only two-thirds focused on non-perennial river systems. Only "arid," "ephemeral," "intermittent," "non-perennial," and "temporary" had over 80% of papers report an epithet definition.

Topics chosen for our definition analysis demonstrate the overlap between our multiple epithets. There was direct overlap between "intermittent," "temporary," and "non-perennial." "Non-permanent" and "seasonal" were also closely related to this cluster. This broad overlap demonstrates that using 12 epithets across non-perennial river research is redundant and could lead to confusion when attempting to synthesize papers for further research.

The summary definitions by broad fields demonstrate overlaps as well as unique characteristics of specific epithets. "Intermittent" and "temporary" demonstrate overlap in their broad definitions. Across all three fields, seasonal or predictable cycles are mentioned as well as a lack of surface water leading to isolated pools. Thus, we conclude that despite the range of fields which use various epithets, there is large overlap in their definitions. We also found that descriptive rather than quantitative definitions (e.g., how many weeks or months a river was without water) were more common throughout our definition analysis. While descriptive definitions have merit, we argue that a lack of quantitative definitions make classifying rivers and streams more difficult. While previous attempts at offering definitions included quantitative definitions [2,17], their attempts at offering universal definitions were unsuccessful. As previously stated, these quantitative definitions were extremely specific, requiring hydrological data that is often unavailable for non-perennial systems. Therefore, we argue that while quantitative definitions are useful in stream classification, they must be broad enough to allow for a lack of hydrological data. We further conclude that the use of so many epithets for non-perennial river systems is more confusing than helpful, highlighting the need for common epithets and definitions.

The results from our study must be interpreted with the following limitations in mind. First, our literature search was designed to be as specific as possible (limited epithets, waterbody terms, and categories), yet papers reporting on research other than non-perennial systems made it into our analyses. Examining each paper individually was outside the scope of this study. While the papers on other topics may skew our results, they are also indicative of the papers a researcher would identify when conducting a WoS search for a meta-analysis. Second, our review is based solely on the WoS search engine. Following similar methods with Google Scholar searches may have resulted in a different set of papers. However, the number of papers returned by our WoS search provided us with a large corpus that we believe represented the body of literature on non-perennial systems. We therefore chose to limit overlap by focusing on WoS search results. These methods may have excluded some papers on non-perennial river systems [1], however the aim of the paper was to capture overall trends in non-perennial river research. We felt that the logic behind our methods was sound and maintaining these methods prevented any biases in data collection.

## 5. Recommendations and Conclusions

We find the use of multiple epithets of non-perennial river research to be redundant and confounding, likely challenging literature syntheses and meta-analyses in the future, and potentially limiting communication and knowledge exchange among researchers. Of the twelve epithets we analyzed, we suggest the continued use of the following three:

1. Non-perennial: this epithet typically had the broadest, but also the fewest, definitions. We speculate the lack of definitions is in part due to the broad nature of the epithet and that there is an implicit assumption that readers understand what the term means. We suggest the following definition: Any lotic, freshwater system that periodically ceases to flow and/or is dry at some point in time and/or space.

2. Intermittent: this epithet frequently overlapped with "temporary" (Figures 2 and 5). We chose "intermittent" as opposed to "temporary" because "temporary river" could imply that the river channel is not always present, as opposed to the surface flow within the riverbed being temporary. We argue that because the riverbed is always present, even if there is no water flowing, use of the word temporary is misleading. Intermittent rivers are those which do not only depend solely on precipitation for surface flow, and interface with groundwater that allows for prolonged flow. We suggest the following definition: A non-perennial river or stream with a considerable connection to the groundwater table, having variable cycles of wetting and flow cessation, and with flow that is sustained longer than a single storm event. These waterways are hydrologically gaining the majority of the time when considering long term flow patterns.

3. Ephemeral: "ephemeral" was not included in the same clusters as "intermittent" (Figures 2 and 5), implying that the two epithets lack overlap among their topics. Definitions for ephemeral were often related to precipitation patterns or extreme events, such as flooding and droughts, although not all definitions included these extremes and not every river that dries during a severe drought is ephemeral. However, the general focus on precipitation events implies that ephemeral streams lack a connection with groundwater and depend solely on precipitation for flow. We suggest the following definition: A type of non-perennial river or stream without a considerable groundwater connection that flows for a short period of time, typically only after precipitation events. These waterways are hydrologically losing the majority of the time when considering long term flow patterns.

These definitions were designed to be flexible enough to encompass existing definitions and enable their use across space and time [21]. We also avoid broader epithets to be more specific for non-perennial systems as well as avoiding epithets that seem to reference perennial flow ("seasonal," "dry," "discontinuous.") In addition, these three epithets are used and defined in the 2020 Navigable Waters Protection Rule by the U.S. EPA. We acknowledge that the wide range of epithets currently used to describe these systems reflects the range of non-perennial systems in space and time as well as various disciplinary and cultural perspectives. We also acknowledge that many scientists may be accustomed to using epithets not listed here and may have their reasons to resist changing their epithet use. We strongly suggest that clear definitions be provided regardless of which epithets researchers use, or that authors cite a reference that provides a clear definition. Ideally, this definition would be located early in the introduction, or by clearly identifying a study site as non-perennial with a quantitative analysis or measurement that underlies its classification. By offering a clear definition, authors will ensure that their work will be included in research surrounding non-perennial river systems, ensuring that it will be included in future syntheses.

The use of consistent epithets for non-perennial research is imperative not only for future research, but also for policies surrounding these river systems [1]. Non-perennial systems are dynamic, variable, and abundant [1–3,6]. They play an important role in ecosystems, and their protection and management are vital to biodiversity and human health. Effectively communicating research using consistent language and providing clear definitions for key terms will enable scientific research to effectively bridge research fields [21], enhance communication with the public [13–15], and influence protections surrounding these systems [22].

**Supplementary Materials:** The following are available online at http://www.mdpi.com/2073-4441/12/7/1980/s1, Figure S1: Methods Flow Chart. Figure S2: Coherence scores over number of topics for Complete Corpus. Figure S3: Coherence scores over number of topics for Time Series Corpus. Figure S4: Time Series topics over time. Table S1: Papers used in definition analysis. Table S2: Definition analysis themes. Table S3: Web of Science categories by research field. To see the code that was used to develop this manuscript, please visit: https://github.com/shelleydunkey/What_R_IRES_text_mining.

**Author Contributions:** Conceptualization, D.C.A., K.H.C., M.H.B., K.S.B., T.D., K.M.F., S.G., J.D.O., M.Z., A.S.W., M.C.M., M.T.B., S.K.K., W.K.D.; methodology, D.C.A., K.H.C., and M.H.B.; software, M.H.B.; validation, D.C.A., K.H.C., and M.H.B.; formal analysis, M.H.B.; investigation, C.A.K., J.C.H., K.H.C., K.M.F., M.P.R., K.S.B., M.S., M.Z., R.M.B., T.D., W.K.D., M.H.B.; resources, D.C.A., K.H.C., M.H.B., K.S.B., T.D., K.F., S.G., J.D.O., M.Z., A.S.W., M.C.M., M.T.B., S.K.K., W.K.D.; data curation, M.H.B.; writing—original draft preparation, M.H.B, K.H.C., D.C.A.; writing—review and editing, K.H.C., D.C.A., C.A.K., T.D., K.F., M.S., M.Z., S.K.K., R.M.B., J.C.H., J.D.O., G.H.A., W.K.D., J.R.B., K.S.B., C.N.J.; visualization, M.H.B., C.A.K.; supervision, D.C.A., K.H.C.; project administration, D.C.A., K.H.C., M.H.B.; funding acquisition, D.C.A., K.H.C. All authors have read and agreed to the published version of the manuscript.

**Funding:** This manuscript is a product of the Dry Rivers Research Coordination Network, which was supported by funding from the US National Science Foundation (DEB-1754389).

**Acknowledgments:** We gratefully acknowledge the help of the University of Oklahoma DAVIS librarians, in particular Sarah Pugachev, for their assistance in coding. We also acknowledge those who helped with the formation of this project: Albert Ruhi and Sarah Godsey. We would like to acknowledge Caryn Vaughn, Tom Neeson, and Katharine Marske for their helpful comments and edits. We also acknowledge the assistance received from undergraduate researchers in the lab of K.H.C.: Gianna N. St. Julien and Alexandra Trahan. Although this work was reviewed by the USEPA, and approved for publication, it does not necessarily reflect official USEPA policy. Any use of trade, firm, or product names is for descriptive purposes only and does not imply endorsement by the U.S. Government.

**Conflicts of Interest:** The authors declare no conflict of interest.

## Appendix A

The six topics from all twelve epithets as identified with latent Dirichlet allocation (LDA) topic modeling from the complete corpus made up of 11,989 total abstracts. We assigned topic names to each topic after looking for similarities among the top twenty words from each topic. The top 20 words are stemmed, or, reduced to their base form.

**Table A1.** Complete Corpus Topics.

| Topic Number | Topic Name | Top 20 Topic Words (Stemmed) |
| --- | --- | --- |
| t_1 | Geomorphology | sediment *, channel *, flow *, water *, river *, deposit *, stream *, groundwat *, concentr *, surfac *, flood *, studi *, area *, season *, lake *, event *, basin *, transport *, process *, discharg * |
| t_2 | Vegetation | speci *, season *, flower *, plant *, burn *, tree *, fire *, popul *, forest *, habitat *, area *, site *, differ *, year *, studi *, veget *, seed *, growth *, us *, dry * |
| t_3 | Ecohydrology | stream *, flow *, river *, speci *, water *, dry *, commun *, fish *, intermitt *, site *, habitat *, season *, chang *, differ *, variabl *, studi *, increas *, us *, hydrolog *, temporari * |
| t_4 | Agriculture | soil *, water *, irrig *, yield *, us *, season *, crop *, increas *, eros *, treatment *, gulli *, plant *, differ *, year *, effect *, studi *, field *, ha *, product *, mm * |
| t_5 | Climate | water *, temperatur *, dry *, season *, rate *, increas *, us *, concentr *, degre *, differ *, leaf *, result *, effect *, flow *, plant *, activ *, cell *, growth *, studi *, co * |
| t_6 | Hydrology | water *, model *, us *, season *, chang *, flow *, data *, river *, streamflow *, climat *, studi *, runoff *, basin *, hydrolog *, result *, forecast *, simul *, watersh *, method *, area * |

## Appendix B

The 9 topics that resulted from LDA topic modeling across all six timeframe corpora. Timeframe corpora were based on WoS searches done after abstracts were common (1990) and broken up into

5-year timeframes (except the final 2016–2018 timeframe). We assigned topic names to each topic based on the top 20 words for each topic.

**Table A2.** Time Series Corpus Topics.

| Topic Number | Topic Name | Top 20 Topic Words (Stemmed) |
|:---:|:---:|:---:|
| t_1 | Community ecology | stream *, water *, river *, flow *, season *, fish *, commun *, site *, dry *, speci *, intermitt *, sampl *, habitat *, concentr, * lake *, assemblag, studi *, differ *, pool *, variabl * |
| t_2 | Hydrology | water *, model *, season *, chang *, river *, flow *, us *, climat *, streamflow *, basin *, forecast *, hydrolog *, increas, precipit *, manag *, region *, temperatur *, variabl *, runoff *, data * |
| t_3 | Water quality | flow *, water *, model *, us *, temperatur *, surfac *, dry *, result *, particl *, condit *, solut *, observ *, differ *, studi *, heat *, measur *, time *, process *, effect *, wave * |
| t_4 | Soil science | soil *, water *, season *, increas *, dry *, emiss *, content *, rate *, us *, moistur *, concentr *, drainag *, root *, potenti *, cm *, measur *, co *, rice *, carbon *, soil water * |
| t_5 | Agriculture | water *, irrig *, yield *, crop *, us *, season *, plant *, treatment *, increas *, stress *, product *, water us *, flower *, growth *, soil *, differ *, leaf *, grain *, effici *, effect * |
| t_6 | Riparian | speci *, season *, forest *, burn *, fire *, veget *, tree *, plant *, site *, area *, habitat *, dry *, riparian *, us *, year *, increas *, effect *, water *, differ *, commun * |
| t_7 | Geomorphology | sediment *, channel *, deposit *, river *, gulli *, flow *, flood *, basin *, eros *, ephemer *, area *, format *, chang *, system *, lake *, bed *, sand *, surfac *, vallei *, studi * |
| t_8 | Modeling | water *, model *, us *, runoff *, groundwat *, soil *, area *, flow *, watersh *, data *, catchment *, studi *, sediment *, eros *, rainfal *, estim *, qualiti *, stream *, land *, result * |
| t_9 | Population ecology | speci *, popul *, flower *, fish *, temperatur *, differ *, habitat *, genet *, studi *, season *, femal *, egg *, rate *, us *, new *, male *, size *, growth *, degre *, reproduct * |

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
