# Peer review of "What’s in a Name? Patterns, Trends, and Suggestions for Defining Non-Perennial Rivers and Streams"

_water, doi:10.3390/w12071980_

Round 1
Reviewer 1 Report
I usually do not pay a lot of attention to the papers analyzing the WoS results. Yet, I found this manuscript interesting. In my opinion the manuscript provides a good unification for naming the non-perennial rivers. I have only very few minor complains, which I list below and of which the most important is lack of statistical analysis. Also the writing style is a bit too poetic, especially in the introduction.
L52, 83…, and elsewhere. Perhaps it would be better to use the term “epithet” or “attribute” instead of “adjective”, because the paper is not about the parts of speech but about the words describing a type river.
L 81. Litany of sth is used to describe a number of unpleasant things (or a Christian prayer). Perhaps ”A number of terms” would be better.
L101. River gauge instead of flow gauge (Which is used in pipes, not in rivers)
L132. Is “ISI Web od Science” still a valid name? I am thinking of the “ISI” abbreviation.
L176. What is the probabilistic coherence of topics? A matix with some measure of similarity? Where the 50 topics came from?
Table 1. Isn’t the term “irregular” more related to the shape of the river (whether it was regulated or not)? If so, this could disrupt the results of the analysis. Perhaps it is worth to check in what context “irregular” appeared in those 347 papers.
Figure 1 is very difficult to synthesize for me. It is very hard to judge which adjective is the most frequently used in which discipline and whether the differences are significant. Instead of this figure I would just show the dominant adjective for each discipline and perform the Kruskal-Wallis test (or anova if the experiment is balanced) to see if the differences are significant. A post-hoc test would also be useful for a deeper analysis, but I do not think it is necessary.
Figure 3. Similarly as in figure 1, a statistical analysis showing the significance of the differences would be very useful. Maybe a Mann-Kendall test can be used here; I am not sure whether it can be used for discrete scales... Otherwise the KW test can help.
Figure 4. This stacked plot is difficult to analyze. A simple line plot where each line has the same origin would show better which adjective appeared in the highest number of papers in each year.
Figure 5. Instead of providing the definitions of the abbreviation in the caption please provide the full names in the figure. There is a plenty of place there.
Section 4.1 and 4.2 The discussion here would be much more meaningful if the statistical test, which were mentioned earlier in the review) were conducted.
Reviewer 2 Report
The manuscript is very well prepared and the presented topic is very interesting. The results are very well described and documented.
Main comments:
- In the methodology section it is worthwhile to present a graphical scheme of the research procedure.
- Please explain what’s mean in Figure 2 – stress value = 0.1029 and in Figure 5 - stress value = 0.0962.
Specific comments:
- Line 93. Change Uys and O’Keeffe to Uys and O’Keeffe [2].
- Line 203. Change tf – idf to tf_idf
- Figure 2. and Figure 5. The description of the X and Y axes should be changed to nMDS1 and nNMS2
- Figure 5. The labels are overlapping.
Reviewer 3 Report
This is a comprehensive review paper on the topic of non-perennial streams by analysing literatures searched from WoS. I found this paper interesting and easy to follow, most importantly, it may help scientist and community to clear the definiation of the non-perennial river system. I would recommend it accepted by the journal.
Although it is not aimed to review papers individually, there still are some papers providing some definitions of non-perennial rivers. It would be further completed if comparison of different comparisons can be added.
